# VAE Learning via Stein Variational Gradient Descent

**Yunchen Pu, Zhe Gan, Ricardo Henao, Chunyuan Li, Shaobo Han, Lawrence Carin**
Department of Electrical and Computer Engineering, Duke University
{yp42, zg27, r.henao, cl319, shaobo.han, lcarin}@duke.edu

## Abstract

A new method for learning variational autoencoders (VAEs) is developed, based on Stein variational gradient descent. A key advantage of this approach is that one need not make parametric assumptions about the form of the encoder distribution. Performance is further enhanced by integrating the proposed encoder with importance sampling. Excellent performance is demonstrated across multiple unsupervised and semi-supervised problems, including semi-supervised analysis of the ImageNet data, demonstrating the scalability of the model to large datasets.

## 1 Introduction

There has been significant recent interest in the variational autoencoder (VAE) [11], a generalization of the original autoencoder [33]. VAEs are typically trained by maximizing a variational lower bound of the data log-likelihood [2, 10, 11, 12, 18, 21, 22, 23, 30, 34, 35]. To compute the variational expression, one must be able to explicitly evaluate the associated distribution of latent features, *i.e.*, the stochastic encoder must have an explicit analytic form. This requirement has motivated design of encoders in which a neural network maps input data to the parameters of a simple distribution, *e.g.*, Gaussian distributions have been widely utilized [1, 11, 27, 25].

The Gaussian assumption may be too restrictive in some cases [28]. Consequently, recent work has considered normalizing flows [28], in which random variables from (for example) a Gaussian distribution are fed through a series of nonlinear functions to increase the complexity and representational power of the encoder. However, because of the need to explicitly evaluate the distribution within the variational expression used when learning, these nonlinear functions must be relatively simple, *e.g.*, planar flows. Further, one may require many layers to achieve the desired representational power.

We present a new approach for training a VAE. We recognize that the need for an explicit form for the encoder distribution is only a consequence of the fact that learning is performed based on the variational lower bound. For inference (*e.g.*, at test time), we *do not* need an explicit form for the distribution of latent features, we only require fast sampling from the encoder. Consequently, rather than directly employing the traditional variational lower bound, we seek to minimize the Kullback-Leibler (KL) distance between the true posterior of model and latent parameters. Learning then becomes a novel application of Stein variational gradient descent (SVGD) [15], constituting its first application to training VAEs. We extend SVGD with importance sampling [1], and also demonstrate its novel use in semi-supervised VAE learning.

The concepts developed here are demonstrated on a wide range of unsupervised and semi-supervised learning problems, including a large-scale semi-supervised analysis of the ImageNet dataset. These experimental results illustrate the advantage of SVGD-based VAE training, relative to traditional approaches. Moreover, the results demonstrate further improvements realized by integrating SVGD with importance sampling.

Independent work by [3, 6] proposed the similar models, in which the aurthers incorporated SVGD with VAEs [3] and importance sampling [6] for unsupervised learning tasks.

## 2 Stein Learning of Variational Autoencoder (Stein VAE)

### 2.1 Review of VAE and Motivation for Use of SVGD

Consider data $\mathcal{D} = \{\boldsymbol{x}_n\}_{n=1}^N$, where $\boldsymbol{x}_n$ are modeled via *decoder* $\boldsymbol{x}_n|\boldsymbol{z}_n \sim p(\boldsymbol{x}|\boldsymbol{z}_n; \boldsymbol{\theta})$. A prior $p(\boldsymbol{z})$ is placed on the latent codes. To learn parameters $\boldsymbol{\theta}$, one typically is interested in maximizing the empirical expected log-likelihood, $\frac{1}{N}\sum_{n=1}^N \log p(\boldsymbol{x}_n; \boldsymbol{\theta})$. A variational lower bound is often employed:

$$\mathcal{L}(\boldsymbol{\theta}, \boldsymbol{\phi}; \boldsymbol{x}) = \mathbb{E}_{\boldsymbol{z}|\boldsymbol{x};\boldsymbol{\phi}} \log \left[ \frac{p(\boldsymbol{x}|\boldsymbol{z}; \boldsymbol{\theta})p(\boldsymbol{z})}{q(\boldsymbol{z}|\boldsymbol{x}; \boldsymbol{\phi})} \right] = -\mathrm{KL}(q(\boldsymbol{z}|\boldsymbol{x}; \boldsymbol{\phi})\|p(\boldsymbol{z}|\boldsymbol{x}; \boldsymbol{\theta})) + \log p(\boldsymbol{x}; \boldsymbol{\theta}) , \quad (1)$$

with $\log p(\boldsymbol{x}; \boldsymbol{\theta}) \geq \mathcal{L}(\boldsymbol{\theta}, \boldsymbol{\phi}; \boldsymbol{x})$, and where $\mathbb{E}_{\boldsymbol{z}|\boldsymbol{x};\boldsymbol{\phi}}[\cdot]$ is approximated by averaging over a finite number of samples drawn from *encoder* $q(\boldsymbol{z}|\boldsymbol{x}; \boldsymbol{\phi})$. Parameters $\boldsymbol{\theta}$ and $\boldsymbol{\phi}$ are typically iteratively optimized via stochastic gradient descent [11], seeking to maximize $\sum_{n=1}^N \mathcal{L}(\boldsymbol{\theta}, \boldsymbol{\phi}; \boldsymbol{x}_n)$.

To evaluate the variational expression in (1), we require the ability to sample efficiently from $q(\boldsymbol{z}|\boldsymbol{x}; \boldsymbol{\phi})$, to approximate the expectation. We also require a closed form for this encoder, to evaluate $\log[p(\boldsymbol{x}|\boldsymbol{z}; \boldsymbol{\theta})p(\boldsymbol{z})/q(\boldsymbol{z}|\boldsymbol{x}; \boldsymbol{\phi})]$. In the proposed VAE learning framework, rather than maximizing the variational lower bound explicitly, we focus on the term $\mathrm{KL}(q(\boldsymbol{z}|\boldsymbol{x}; \boldsymbol{\phi})\|p(\boldsymbol{z}|\boldsymbol{x}; \boldsymbol{\theta}))$, which we seek to minimize. This can be achieved by leveraging Stein variational gradient descent (SVGD) [15]. Importantly, for SVGD we need only be able to sample from $q(\boldsymbol{z}|\boldsymbol{x}; \boldsymbol{\phi})$, and we need not possess its explicit functional form.

In the above discussion, $\boldsymbol{\theta}$ is treated as a *parameter*; below we treat it as a *random variable*, as was considered in the Appendix of [11]. Treatment of $\boldsymbol{\theta}$ as a random variable allows for model averaging, and a point estimate of $\boldsymbol{\theta}$ is revealed as a special case of the proposed method.

The set of codes associated with all $\boldsymbol{x}_n \in \mathcal{D}$ is represented $\mathcal{Z} = \{\boldsymbol{z}_n\}_{n=1}^N$. The prior on $\{\boldsymbol{\theta}, \mathcal{Z}\}$ is here represented as $p(\boldsymbol{\theta}, \mathcal{Z}) = p(\boldsymbol{\theta})\prod_{n=1}^N p(\boldsymbol{z}_n)$. We desire the posterior $p(\boldsymbol{\theta}, \mathcal{Z}|\mathcal{D})$. Consider the *revised* variational expression

$$\mathcal{L}_1(q; \mathcal{D}) = \mathbb{E}_{q(\boldsymbol{\theta}, \mathcal{Z})} \log \left[ \frac{p(\mathcal{D}|\mathcal{Z}, \boldsymbol{\theta})p(\boldsymbol{\theta}, \mathcal{Z})}{q(\boldsymbol{\theta}, \mathcal{Z})} \right] = -\mathrm{KL}(q(\boldsymbol{\theta}, \mathcal{Z})\|p(\boldsymbol{\theta}, \mathcal{Z}|\mathcal{D})) + \log p(\mathcal{D}; \mathcal{M}) , \quad (2)$$

where $p(\mathcal{D}; \mathcal{M})$ is the evidence for the underlying model $\mathcal{M}$. Learning $q(\boldsymbol{\theta}, \mathcal{Z})$ such that $\mathcal{L}_1$ is maximized is equivalent to seeking $q(\boldsymbol{\theta}, \mathcal{Z})$ that minimizes $\mathrm{KL}(q(\boldsymbol{\theta}, \mathcal{Z})\|p(\boldsymbol{\theta}, \mathcal{Z}|\mathcal{D}))$. By leveraging and generalizing SVGD, we will perform the latter.

### 2.2 Stein Variational Gradient Descent (SVGD)

Rather than explicitly specifying a form for $p(\boldsymbol{\theta}, \mathcal{Z}|\mathcal{D})$, we sequentially refine samples of $\boldsymbol{\theta}$ and $\mathcal{Z}$, such that they are better matched to $p(\boldsymbol{\theta}, \mathcal{Z}|\mathcal{D})$. We alternate between updating the samples of $\boldsymbol{\theta}$ and samples of $\mathcal{Z}$, analogous to how $\boldsymbol{\theta}$ and $\boldsymbol{\phi}$ are updated alternatively in traditional VAE optimization of (1). We first consider updating samples of $\boldsymbol{\theta}$, with the samples of $\mathcal{Z}$ held fixed. Specifically, assume we have samples $\{\boldsymbol{\theta}_j\}_{j=1}^M$ drawn from distribution $q(\boldsymbol{\theta})$, and samples $\{\boldsymbol{z}_{jn}\}_{j=1}^M$ drawn from distribution $q(\mathcal{Z})$. We wish to transform $\{\boldsymbol{\theta}_j\}_{j=1}^M$ by feeding them through a function, and the corresponding (implicit) transformed distribution from which they are drawn is denoted as $q_T(\boldsymbol{\theta})$. It is desired that, in a KL sense, $q_T(\boldsymbol{\theta})q(\mathcal{Z})$ is closer to $p(\boldsymbol{\theta}, \mathcal{Z}|\mathcal{D})$ than was $q(\boldsymbol{\theta})q(\mathcal{Z})$. The following theorem is useful for defining how to best update $\{\boldsymbol{\theta}_j\}_{j=1}^M$.

**Theorem 1** *Assume $\boldsymbol{\theta}$ and $\mathcal{Z}$ are Random Variables (RVs) drawn from distributions $q(\boldsymbol{\theta})$ and $q(\mathcal{Z})$, respectively. Consider the transformation $T(\boldsymbol{\theta}) = \boldsymbol{\theta} + \epsilon\psi(\boldsymbol{\theta}; \mathcal{D})$ and let $q_T(\boldsymbol{\theta})$ represent the distribution of $\boldsymbol{\theta}' = T(\boldsymbol{\theta})$. We have*

$$\nabla_\epsilon \Big( KL(q_T\|p) \Big)\big|_{\epsilon=0} = -\mathbb{E}_{\boldsymbol{\theta} \sim q(\boldsymbol{\theta})} \big( \mathrm{trace}(\mathcal{A}_p(\boldsymbol{\theta}; \mathcal{D})) \big) , \quad (3)$$

*where $q_T = q_T(\boldsymbol{\theta})q(\mathcal{Z})$, $p = p(\boldsymbol{\theta}, \mathcal{Z}|\mathcal{D})$, $\mathcal{A}_p(\boldsymbol{\theta}; \mathcal{D}) = \nabla_{\boldsymbol{\theta}} \log \tilde{p}(\boldsymbol{\theta}; \mathcal{D})\psi(\boldsymbol{\theta}; \mathcal{D})^T + \nabla_{\boldsymbol{\theta}}\psi(\boldsymbol{\theta}; \mathcal{D})$, $\log \tilde{p}(\boldsymbol{\theta}; \mathcal{D}) = \mathbb{E}_{\mathcal{Z} \sim q(\mathcal{Z})}[\log p(\mathcal{D}, \mathcal{Z}, \boldsymbol{\theta})]$, and $p(\mathcal{D}, \mathcal{Z}, \boldsymbol{\theta}) = p(\mathcal{D}|\mathcal{Z}, \boldsymbol{\theta})p(\boldsymbol{\theta}, \mathcal{Z})$.*

The proof is provided in Appendix A. Following [15], we assume $\psi(\boldsymbol{\theta}; \mathcal{D})$ lives in a reproducing kernel Hilbert space (RKHS) with kernel $k(\cdot, \cdot)$. Under this assumption, the solution for $\psi(\boldsymbol{\theta}; \mathcal{D})$

that maximizes the decrease in the KL distance (3) is

$$\psi^*(\cdot; \mathcal{D}) = \mathbb{E}_{q(\boldsymbol{\theta})}[k(\boldsymbol{\theta}, \cdot)\nabla_{\boldsymbol{\theta}} \log \tilde{p}(\boldsymbol{\theta}; \mathcal{D}) + \nabla_{\boldsymbol{\theta}} k(\boldsymbol{\theta}, \cdot)] . \tag{4}$$

Theorem 1 concerns updating samples from $q(\boldsymbol{\theta})$ assuming fixed $q(\mathcal{Z})$. Similarly, to update $q(\mathcal{Z})$ with $q(\boldsymbol{\theta})$ fixed, we employ a complementary form of Theorem 1 (omitted for brevity). In that case, we consider transformation $T(\mathcal{Z}) = \mathcal{Z} + \epsilon\psi(\mathcal{Z}; \mathcal{D})$, with $\mathcal{Z} \sim q(\mathcal{Z})$, and function $\psi(\mathcal{Z}; \mathcal{D})$ is also assumed to be in a RKHS.

The expectations in (3) and (4) are approximated by samples $\boldsymbol{\theta}_j^{(t+1)} = \boldsymbol{\theta}_j^{(t)} + \epsilon\Delta\boldsymbol{\theta}_j^{(t)}$, with

$$\Delta\boldsymbol{\theta}_j^{(t)} \approx \tfrac{1}{M}\sum_{j'=1}^M \left[ k_{\boldsymbol{\theta}}(\boldsymbol{\theta}_{j'}^{(t)}, \boldsymbol{\theta}_j^{(t)})\nabla_{\boldsymbol{\theta}_{j'}^{(t)}} \log \tilde{p}(\boldsymbol{\theta}_{j'}^{(t)}; \mathcal{D}) + \nabla_{\boldsymbol{\theta}_{j'}^{(t)}} k_{\boldsymbol{\theta}}(\boldsymbol{\theta}_{j'}^{(t)}, \boldsymbol{\theta}_j^{(t)})) \right] , \tag{5}$$

with $\nabla_{\boldsymbol{\theta}} \log \tilde{p}(\boldsymbol{\theta}; \mathcal{D}) \approx \tfrac{1}{M}\sum_{n=1}^N \sum_{j=1}^M \nabla_{\boldsymbol{\theta}} \log p(\boldsymbol{x}_n|\boldsymbol{z}_{jn}, \boldsymbol{\theta})p(\boldsymbol{\theta})$. A similar update of samples is manifested for the latent variables $\boldsymbol{z}_{jn}^{(t+1)} = \boldsymbol{z}_{jn}^{(t)} + \epsilon\Delta\boldsymbol{z}_{jn}^{(t)}$:

$$\Delta\boldsymbol{z}_{jn}^{(t)} = \tfrac{1}{M}\sum_{j'=1}^M \left[ k_{\boldsymbol{z}}(\boldsymbol{z}_{j'n}^{(t)}, \boldsymbol{z}_{jn}^{(t)})\nabla_{\boldsymbol{z}_{j'n}^{(t)}} \log \tilde{p}(\boldsymbol{z}_{j'n}^{(t)}; \mathcal{D}) + \nabla_{\boldsymbol{z}_{j'n}^{(t)}} k_{\boldsymbol{z}}(\boldsymbol{z}_{j'n}^{(t)}, \boldsymbol{z}_{jn}^{(t)}) \right] , \tag{6}$$

where $\nabla_{\boldsymbol{z}_n} \log \tilde{p}(\boldsymbol{z}_n; \mathcal{D}) \approx \tfrac{1}{M}\sum_{j=1}^M \nabla_{\boldsymbol{z}_n} \log p(\boldsymbol{x}_n|\boldsymbol{z}_n, \boldsymbol{\theta}_j')p(\boldsymbol{z}_n)$. The kernels used to update samples of $\boldsymbol{\theta}$ and $\boldsymbol{z}_n$ are in general different, denoted respectively $k_{\boldsymbol{\theta}}(\cdot, \cdot)$ and $k_{\boldsymbol{z}}(\cdot, \cdot)$, and $\epsilon$ is a small step size. For notational simplicity, $M$ is the same in (5) and (6), but in practice a different number of samples may be used for $\boldsymbol{\theta}$ and $\mathcal{Z}$.

If $M = 1$ for parameter $\boldsymbol{\theta}$, indices $j$ and $j'$ are removed in (5). Learning then reduces to gradient descent and a point estimate for $\boldsymbol{\theta}$, identical to the optimization procedure used for the traditional VAE expression in (1), but with the (multiple) samples associated with $\mathcal{Z}$ sequentially transformed via SVGD (and, importantly, without the need to assume a form for $q(\boldsymbol{z}|\boldsymbol{x}; \boldsymbol{\phi})$). Therefore, if only a point estimate of $\boldsymbol{\theta}$ is desired, (1) can be optimized wrt $\boldsymbol{\theta}$, while for updating $\mathcal{Z}$ SVGD is applied.

## 2.3 Efficient Stochastic Encoder

At iteration $t$ of the above learning procedure, we realize a set of latent-variable (code) samples $\{\boldsymbol{z}_{jn}^{(t)}\}_{j=1}^M$ for each $\boldsymbol{x}_n \in \mathcal{D}$ under analysis. For large $N$, training may be computationally expensive. Further, the need to evolve (learn) samples $\{\boldsymbol{z}_{j*}\}_{j=1}^M$ for each new test sample, $\boldsymbol{x}_*$, is undesirable. We therefore develop a *recognition model* that efficiently computes samples of latent codes for a data sample of interest. The recognition model draws samples via $\boldsymbol{z}_{jn} = \boldsymbol{f}_{\boldsymbol{\eta}}(\boldsymbol{x}_n, \boldsymbol{\xi}_{jn})$ with $\boldsymbol{\xi}_{jn} \sim q_0(\boldsymbol{\xi})$. Distribution $q_0(\boldsymbol{\xi})$ is selected such that it may be easily sampled, *e.g.*, isotropic Gaussian.

After each iteration of updating the samples of $\mathcal{Z}$, we refine recognition model $\boldsymbol{f}_{\boldsymbol{\eta}}(\boldsymbol{x}, \boldsymbol{\xi})$ to mimic the Stein sample dynamics. Assume recognition-model parameters $\boldsymbol{\eta}^{(t)}$ have been learned thus far. Using $\boldsymbol{\eta}^{(t)}$, latent codes for iteration $t$ are constituted as $\boldsymbol{z}_{jn}^{(t)} = \boldsymbol{f}_{\boldsymbol{\eta}^{(t)}}(\boldsymbol{x}_n, \boldsymbol{\xi}_{jn})$, with $\boldsymbol{\xi}_{jn} \sim q_0(\boldsymbol{\xi})$. These codes are computed for all data $\boldsymbol{x}_n \in \mathcal{B}_t$, where $\mathcal{B}_t \subset \mathcal{D}$ is the minibatch of data at iteration $t$. The change in the codes is $\Delta\boldsymbol{z}_{jn}^{(t)}$, as defined in (6). We then update $\boldsymbol{\eta}$ to match the refined codes, as

$$\boldsymbol{\eta}^{(t+1)} = \arg\min_{\boldsymbol{\eta}} \sum_{\boldsymbol{x}_n \in \mathcal{B}_t} \sum_{j=1}^M \|\boldsymbol{f}_{\boldsymbol{\eta}}(\boldsymbol{x}_n, \boldsymbol{\xi}_{jn}) - \boldsymbol{z}_{jn}^{(t+1)}\|^2 . \tag{7}$$

The analytic solution of (7) is intractable. We update $\boldsymbol{\eta}$ with $K$ steps of gradient descent as $\boldsymbol{\eta}^{(t,k)} = \boldsymbol{\eta}^{(t,k-1)} - \delta\sum_{\boldsymbol{x}_n \in \mathcal{B}_t} \sum_{j=1}^M \Delta\boldsymbol{\eta}_{jn}^{(t,k-1)}$, where $\Delta\boldsymbol{\eta}_{jn}^{(t,k-1)} = \partial_{\boldsymbol{\eta}}\boldsymbol{f}_{\boldsymbol{\eta}}(\boldsymbol{x}_n, \boldsymbol{\xi}_{jn})(\boldsymbol{f}_{\boldsymbol{\eta}}(\boldsymbol{x}_n, \boldsymbol{\xi}_{jn}) - \boldsymbol{z}_{jn}^{(t+1)})|_{\boldsymbol{\eta}=\boldsymbol{\eta}^{(t,k-1)}}$, $\delta$ is a small step size, $\boldsymbol{\eta}^{(t)} = \boldsymbol{\eta}^{(t,0)}$, $\boldsymbol{\eta}^{(t+1)} = \boldsymbol{\eta}^{(t,K)}$, and $\partial_{\boldsymbol{\eta}}\boldsymbol{f}_{\boldsymbol{\eta}}(\boldsymbol{x}_n, \boldsymbol{\xi}_{jn})$ is the transpose of the Jacobian of $\boldsymbol{f}_{\boldsymbol{\eta}}(\boldsymbol{x}_n, \boldsymbol{\xi}_{jn})$ wrt $\boldsymbol{\eta}$. Note that the use of minibatches mitigates challenges of training with large training sets, $\mathcal{D}$.

The function $\boldsymbol{f}_{\boldsymbol{\eta}}(\boldsymbol{x}, \boldsymbol{\xi})$ plays a role analogous to $q(\boldsymbol{z}|\boldsymbol{x}; \boldsymbol{\phi})$ in (1), in that it yields a means of efficiently drawing samples of latent codes $\boldsymbol{z}$, given observed $\boldsymbol{x}$; however, we do not impose an explicit functional form for the distribution of these samples.

# 3 Stein Variational Importance Weighted Autoencoder (Stein VIWAE)

## 3.1 Multi-sample importance-weighted KL divergence

Recall the variational expression in (1) employed in conventional VAE learning. Recently, [1, 19] showed that the multi-sample ($k$ samples) importance-weighted estimator

$$\mathcal{L}_k(\boldsymbol{x}) = \mathbb{E}_{\boldsymbol{z}^1,\ldots,\boldsymbol{z}^k \sim q(\boldsymbol{z}|\boldsymbol{x})} \left[ \log \frac{1}{k} \sum_{i=1}^k \frac{p(\boldsymbol{x},\boldsymbol{z}^i)}{q(\boldsymbol{z}^i|\boldsymbol{x})} \right], \tag{8}$$

provides a tighter lower bound and a better proxy for the log-likelihood, where $\boldsymbol{z}^1,\ldots,\boldsymbol{z}^k$ are random variables sampled independently from $q(\boldsymbol{z}|\boldsymbol{x})$. Recall from (3) that the KL divergence played a key role in the Stein-based learning of Section 2. Equation (8) motivates replacement of the KL objective function with the multi-sample importance-weighted KL divergence

$$\mathrm{KL}_{q,p}^k(\boldsymbol{\Theta}; \mathcal{D}) \triangleq -\mathbb{E}_{\boldsymbol{\Theta}^{1:k} \sim q(\boldsymbol{\Theta})} \left[ \log \frac{1}{k} \sum_{i=1}^k \frac{p(\boldsymbol{\Theta}^i|\mathcal{D})}{q(\boldsymbol{\Theta}^i)} \right], \tag{9}$$

where $\boldsymbol{\Theta} = (\boldsymbol{\theta}, \mathcal{Z})$ and $\boldsymbol{\Theta}^{1:k} = \boldsymbol{\Theta}^1, \ldots, \boldsymbol{\Theta}^k$ are independent samples from $q(\boldsymbol{\theta}, \mathcal{Z})$. Note that the special case of $k = 1$ recovers the standard KL divergence. Inspired by [1], the following theorem (proved in Appendix A) shows that increasing the number of samples $k$ is guaranteed to reduce the KL divergence and provide a better approximation of target distribution.

**Theorem 2** *For any natural number $k$, we have $KL_{q,p}^k(\boldsymbol{\Theta}; \mathcal{D}) \geq KL_{q,p}^{k+1}(\boldsymbol{\Theta}; \mathcal{D}) \geq 0$, and if $q(\boldsymbol{\Theta})/p(\boldsymbol{\Theta}|\mathcal{D})$ is bounded, then $\lim_{k\to\infty} KL_{q,p}^k(\boldsymbol{\Theta}; \mathcal{D}) = 0$.*

We minimize (9) with a sample transformation based on a generalization of SVGD and the recognition model (encoder) is trained in the same way as in Section 2.3. Specifically, we first draw samples $\{\boldsymbol{\theta}_j^{1:k}\}_{j=1}^M$ and $\{\boldsymbol{z}_{jn}^{1:k}\}_{j=1}^M$ from a simple distribution $q_0(\cdot)$, and convert these to approximate draws from $p(\boldsymbol{\theta}^{1:k}, \mathcal{Z}^{1:k}|\mathcal{D})$ by minimizing the multi-sample importance weighted KL divergence via nonlinear functional transformation.

## 3.2 Importance-weighted SVGD for VAEs

The following theorem generalizes Theorem 1 to multi-sample weighted KL divergence.

**Theorem 3** *Let $\boldsymbol{\Theta}^{1:k}$ be RVs drawn independently from distribution $q(\boldsymbol{\Theta})$ and $KL_{q,p}^k(\boldsymbol{\Theta}, \mathcal{D})$ is the multi-sample importance weighted KL divergence in (9). Let $T(\boldsymbol{\Theta}) = \boldsymbol{\Theta} + \epsilon\psi(\boldsymbol{\Theta}; \mathcal{D})$ and $q_T(\boldsymbol{\Theta})$ represent the distribution of $\boldsymbol{\Theta}' = T(\boldsymbol{\Theta})$. We have*

$$\nabla_\epsilon \left( KL_{q,p}^k(\boldsymbol{\Theta}'; \mathcal{D}) \right)|_{\epsilon=0} = -\mathbb{E}_{\boldsymbol{\Theta}^{1:k} \sim q(\boldsymbol{\Theta})} (\mathcal{A}_p^k(\boldsymbol{\Theta}^{1:k}; \mathcal{D})). \tag{10}$$

The proof and detailed definition is provided in Appendix A. The following corollaries generalize Theorem 1 and (4) via use of importance sampling, respectively.

**Corollary 3.1** *$\boldsymbol{\theta}^{1:k}$ and $\mathcal{Z}^{1:k}$ are RVs drawn independently from distributions $q(\boldsymbol{\theta})$ and $q(\mathcal{Z})$, respectively. Let $T(\boldsymbol{\theta}) = \boldsymbol{\theta} + \epsilon\psi(\boldsymbol{\theta}; \mathcal{D})$, $q_T(\boldsymbol{\theta})$ represent the distribution of $\boldsymbol{\theta}' = T(\boldsymbol{\theta})$, and $\boldsymbol{\Theta}' = (\boldsymbol{\theta}', \mathcal{Z})$. We have*

$$\nabla_\epsilon \left( KL_{q_T,p}^k(\boldsymbol{\Theta}'; \mathcal{D}) \right)|_{\epsilon=0} = -\mathbb{E}_{\boldsymbol{\theta}^{1:k} \sim q(\boldsymbol{\theta})} (\mathcal{A}_p^k(\boldsymbol{\theta}^{1:k}; \mathcal{D})), \tag{11}$$

*where $\mathcal{A}_p^k(\boldsymbol{\theta}^{1:k}; \mathcal{D}) = \frac{1}{\tilde{\omega}} \sum_{i=1}^k \omega_i \mathcal{A}_p(\boldsymbol{\theta}^i; \mathcal{D})$, $\omega_i = \mathbb{E}_{\mathcal{Z}^i \sim q(\mathcal{Z})} \left[ \frac{p(\boldsymbol{\theta}^i, \mathcal{Z}^i, \mathcal{D})}{q(\boldsymbol{\theta}^i)q(\mathcal{Z}^i)} \right]$, $\tilde{\omega} = \sum_{i=1}^k \omega_i$; $\mathcal{A}_p(\boldsymbol{\theta}; \mathcal{D})$ and $\log \tilde{p}(\boldsymbol{\theta}; \mathcal{D})$ are as defined in Theorem 1.*

**Corollary 3.2** *Assume $\psi(\boldsymbol{\theta}; \mathcal{D})$ lives in a reproducing kernel Hilbert space (RKHS) with kernel $k_{\boldsymbol{\theta}}(\cdot, \cdot)$. The solution for $\psi(\boldsymbol{\theta}; \mathcal{D})$ that maximizes the decrease in the KL distance (11) is*

$$\psi^*(\cdot; \mathcal{D}) = \mathbb{E}_{\boldsymbol{\theta}^{1:k} \sim q(\boldsymbol{\theta})} \left[ \frac{1}{\tilde{\omega}} \sum_{i=1}^k \omega_i \left( \nabla_{\boldsymbol{\theta}^i} k_{\boldsymbol{\theta}}(\boldsymbol{\theta}^i, \cdot) + k_{\boldsymbol{\theta}}(\boldsymbol{\theta}^i, \cdot) \nabla_{\boldsymbol{\theta}^i} \log \tilde{p}(\boldsymbol{\theta}^i; \mathcal{D}) \right) \right]. \tag{12}$$

Corollary 3.1 and Corollary 3.2 provide a means of updating multiple samples $\{\boldsymbol{\theta}_j^{1:k}\}_{j=1}^M$ from $q(\boldsymbol{\theta})$ via $T(\boldsymbol{\theta}^i) = \boldsymbol{\theta}^i + \epsilon\psi(\boldsymbol{\theta}^i; \mathcal{D})$. The expectation wrt $q(\mathcal{Z})$ is approximated via samples drawn from $q(\mathcal{Z})$. Similarly, we can employ a complementary form of Corollary 3.1 and Corollary 3.2 to update multiple samples $\{\mathcal{Z}_j^{1:k}\}_{j=1}^M$ from $q(\mathcal{Z})$. This suggests an importance-weighted learning procedure that alternates between update of particles $\{\boldsymbol{\theta}_j^{1:k}\}_{j=1}^M$ and $\{\mathcal{Z}_j^{1:k}\}_{j=1}^M$, which is similar to the one in Section 2.2. Detailed update equations are provided in Appendix B.

## 4 Semi-Supervised Learning with Stein VAE

Consider labeled data as pairs $\mathcal{D}_l = \{\boldsymbol{x}_n, \boldsymbol{y}_n\}_{n=1}^{N_l}$, where the label $\boldsymbol{y}_n \in \{1, \ldots, C\}$ and the decoder is modeled as $(\boldsymbol{x}_n, \boldsymbol{y}_n|\boldsymbol{z}_n) \sim p(\boldsymbol{x}, \boldsymbol{y}|\boldsymbol{z}_n; \boldsymbol{\theta}, \tilde{\boldsymbol{\theta}}) = p(\boldsymbol{x}|\boldsymbol{z}_n; \boldsymbol{\theta})p(\boldsymbol{y}|\boldsymbol{z}_n; \tilde{\boldsymbol{\theta}})$, where $\tilde{\boldsymbol{\theta}}$ represents the parameters of the decoder for labels. The set of codes associated with all labeled data are represented as $\mathcal{Z}_l = \{\boldsymbol{z}_n\}_{n=1}^{N_l}$. We desire to approximate the posterior distribution on the entire dataset $p(\boldsymbol{\theta}, \tilde{\boldsymbol{\theta}}, \mathcal{Z}, \mathcal{Z}_l|\mathcal{D}, \mathcal{D}_l)$ via samples, where $\mathcal{D}$ represents the unlabeled data, and $\mathcal{Z}$ is the set of codes associated with $\mathcal{D}$. In the following, we will only discuss how to update the samples of $\boldsymbol{\theta}$, $\tilde{\boldsymbol{\theta}}$ and $\mathcal{Z}_l$. Updating samples $\mathcal{Z}$ is the same as discussed in Sections 2 and 3.2 for Stein VAE and Stein VIWAE, respectively.

Assume $\{\boldsymbol{\theta}_j\}_{j=1}^M$ drawn from distribution $q(\boldsymbol{\theta})$, $\{\tilde{\boldsymbol{\theta}}_j\}_{j=1}^M$ drawn from distribution $q(\tilde{\boldsymbol{\theta}})$, and samples $\{\boldsymbol{z}_{jn}\}_{j=1}^M$ drawn from (distinct) distribution $q(\mathcal{Z}_l)$. The following corollary generalizes Theorem 1 and (4), which is useful for defining how to best update $\{\boldsymbol{\theta}_j\}_{j=1}^M$.

**Corollary 3.3** *Assume $\boldsymbol{\theta}$, $\tilde{\boldsymbol{\theta}}$, $\mathcal{Z}$ and $\mathcal{Z}_l$ are RVs drawn from distributions $q(\boldsymbol{\theta})$, $q(\tilde{\boldsymbol{\theta}})$, $q(\mathcal{Z})$ and $q(\mathcal{Z}_l)$, respectively. Consider the transformation $T(\boldsymbol{\theta}) = \boldsymbol{\theta} + \epsilon\psi(\boldsymbol{\theta}; \mathcal{D}, \mathcal{D}_l)$ where $\psi(\boldsymbol{\theta}; \mathcal{D}, \mathcal{D}_l)$ lives in a RKHS with kernel $k_{\boldsymbol{\theta}}(\cdot, \cdot)$. Let $q_T(\boldsymbol{\theta})$ represent the distribution of $\boldsymbol{\theta}' = T(\boldsymbol{\theta})$. For $q_T = q_T(\boldsymbol{\theta})q(\mathcal{Z})q(\tilde{\boldsymbol{\theta}})$ and $p = p(\boldsymbol{\theta}, \tilde{\boldsymbol{\theta}}, \mathcal{Z}|\mathcal{D}, \mathcal{D}_l)$, we have*

$$\nabla_\epsilon \Big(KL(q_T\|p)\Big)|_{\epsilon=0} = -\mathbb{E}_{\boldsymbol{\theta}\sim q(\boldsymbol{\theta})}(\mathcal{A}_p(\boldsymbol{\theta}; \mathcal{D}, \mathcal{D}_l)), \qquad (13)$$

*where $\mathcal{A}_p(\boldsymbol{\theta}; \mathcal{D}, \mathcal{D}_l) = \nabla_{\boldsymbol{\theta}}\psi(\boldsymbol{\theta}; \mathcal{D}, \mathcal{D}_l) + \nabla_{\boldsymbol{\theta}}\log\tilde{p}(\boldsymbol{\theta}; \mathcal{D}, \mathcal{D}_l)\psi(\boldsymbol{\theta}; \mathcal{D}, \mathcal{D}_l)^T$, $\log\tilde{p}(\boldsymbol{\theta}; \mathcal{D}, \mathcal{D}_l) = \mathbb{E}_{\mathcal{Z}\sim q(\mathcal{Z})}[\log p(\mathcal{D}|\mathcal{Z}, \boldsymbol{\theta})] + \mathbb{E}_{\mathcal{Z}_l\sim q(\mathcal{Z}_l)}[\log p(\mathcal{D}_l|\mathcal{Z}_l, \boldsymbol{\theta})]$, and the solution for $\psi(\boldsymbol{\theta}; \mathcal{D}, \mathcal{D}_l)$ that maximizes the change in the KL distance (13) is*

$$\psi^*(\cdot; \mathcal{D}, \mathcal{D}_l) = \mathbb{E}_{q(\boldsymbol{\theta})}[k(\boldsymbol{\theta}, \cdot)\nabla_{\boldsymbol{\theta}}\log\tilde{p}(\boldsymbol{\theta}; \mathcal{D}, \mathcal{D}_l) + \nabla_{\boldsymbol{\theta}}k(\boldsymbol{\theta}, \cdot)]. \qquad (14)$$

Further details are provided in Appendix C.

## 5 Experiments

For all experiments, we use a radial basis-function (RBF) kernel as in [15], *i.e.*, $k(\boldsymbol{x}, \boldsymbol{x}') = \exp(-\frac{1}{h}\|\boldsymbol{x} - \boldsymbol{x}'\|_2^2)$, where the bandwidth, $h$, is the median of pairwise distances between current samples. $q_0(\boldsymbol{\theta})$ and $q_0(\boldsymbol{\xi})$ are set to isotropic Gaussian distributions. We share the samples of $\boldsymbol{\xi}$ across data points, *i.e.*, $\boldsymbol{\xi}_{jn} = \boldsymbol{\xi}_j$, for $n = 1, \ldots, N$ (this is not necessary, but it saves computation). The samples of $\boldsymbol{\theta}$ and $\boldsymbol{z}$, and parameters of the recognition model, $\boldsymbol{\eta}$, are optimized via Adam [9] with learning rate 0.0002. We do not perform any dataset-specific tuning or regularization other than dropout [32] and early stopping on validation sets. We set $M = 100$ and $k = 50$, and use minibatches of size 64 for all experiments, unless otherwise specified.

### 5.1 Expressive power of Stein recognition model

**Gaussian Mixture Model** We synthesize data by $(i)$ drawing $\boldsymbol{z}_n \sim \frac{1}{2}\mathcal{N}(\boldsymbol{\mu}_1, \mathbf{I}) + \frac{1}{2}\mathcal{N}(\boldsymbol{\mu}_2, \mathbf{I})$, where $\boldsymbol{\mu}_1 = [5, 5]^T$, $\boldsymbol{\mu}_2 = [-5, -5]^T$; $(ii)$ drawing $\boldsymbol{x}_n \sim \mathcal{N}(\boldsymbol{\theta}\boldsymbol{z}_n, \sigma^2\mathbf{I})$, where $\boldsymbol{\theta} = \begin{bmatrix} 2 & -1 \\ 1 & -2 \end{bmatrix}$ and $\sigma = 0.1$. The recognition model $f_{\boldsymbol{\eta}}(\boldsymbol{x}_n, \boldsymbol{\xi}_j)$ is specified as a multi-layer perceptron (MLP) with 100 hidden units, by first concatenating $\boldsymbol{\xi}_j$ and $\boldsymbol{x}_n$ into a long vector. The dimension of $\boldsymbol{\xi}_j$ is set to 2. The recognition model for standard VAE is also an MLP with 100 hidden units, and with the assumption of a Gaussian distribution for the latent codes [11].

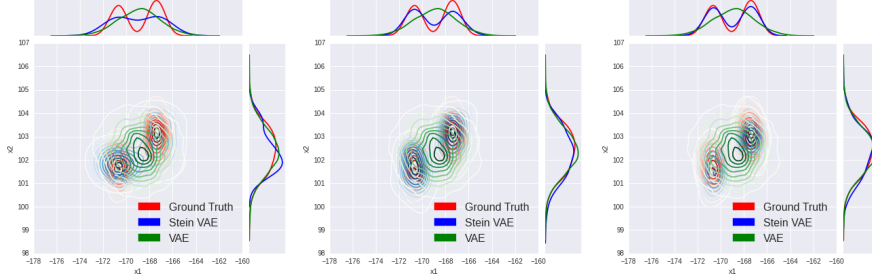

Figure 1: Approximation of posterior distribution: Stein VAE *vs.* VAE. The figures represent different samples of Stein VAE. (left) 10 samples, (center) 50 samples, and (right) 100 samples.

We generate $N = 10,000$ data points for training and 10 data points for testing. The *analytic* form of true posterior distribution is provided in Appendix D. Figure 1 shows the performance of Stein VAE approximations for the true posterior; other similar examples are provided in Appendix F. The Stein recognition model is able to capture the multi-modal posterior and produce accurate density approximation.

**Poisson Factor Analysis** Given a discrete vector $\boldsymbol{x}_n \in \mathbb{Z}_+^P$, Poisson factor analysis [36] assumes $\boldsymbol{x}_n$ is a weighted combination of $V$ latent factors $\boldsymbol{x}_n \sim$ Pois$(\boldsymbol{\theta z}_n)$, where $\boldsymbol{\theta} \in \mathbb{R}_+^{P \times V}$ is the factor loadings matrix and $\boldsymbol{z}_n \in \mathbb{R}_+^V$ is the vector of factor scores. We consider topic modeling with Dirichlet priors on $\boldsymbol{\theta}_v$ ($v$-th column of $\boldsymbol{\theta}$) and gamma priors on each component of $\boldsymbol{z}_n$.

We evaluate our model on the *20 Newsgroups* dataset containing $N = 18,845$ documents with a vocabulary of $P = 2,000$. The data are partitioned into 10,314 training, 1,000 validation and 7,531 test documents. The number of factors (topics) is set to $V = 128$. $\boldsymbol{\theta}$ is first learned by Markov chain Monte Carlo (MCMC) [4]. We then fix $\boldsymbol{\theta}$ at its MAP value, and only learn the recognition model $\boldsymbol{\eta}$ using standard VAE and Stein VAE; this is done, as in the previous example, to examine the accuracy of the recognition model to estimate the posterior of the latent factors, isolated from estimation of $\boldsymbol{\theta}$. The recognition model is an MLP with 100 hidden units.



Figure 2: Univariate marginals and pairwise posteriors. Purple, red and green represent the distribution inferred from MCMC, standard VAE and Stein VAE, respectively.

An analytic form of the true posterior distribution $p(\boldsymbol{z}_n|\boldsymbol{x}_n)$ is intractable for this problem. Consequently, we employ samples collected from MCMC as ground truth. With $\boldsymbol{\theta}$ fixed, we sample $\boldsymbol{z}_n$ via Gibbs sampling, using 2,000 burn-in iterations followed by 2,500 collection draws, retaining every 10th collection sample. We show the marginal and pairwise posterior of one test data point in Figure 2. Additional results are provided in Appendix F. Stein VAE leads to a more accurate approximation than standard VAE, compared to the MCMC samples. Considering Figure 2, note that VAE significantly underestimates the variance of the posterior (examining the marginals), a well-known problem of variational Bayesian analysis [7]. In sharp contrast, Stein VAE yields highly accurate approximations to the true posterior.

Table 1: Negative log-likelihood (NLL) on MNIST. [†]Trained with VAE and tested with IWAE. [‡]Trained and tested with IWAE.

| Method | NLL |
|---|---|
| DGLM [27] | 89.90 |
| Normalizing flow [28] | 85.10 |
| VAE + IWAE [1][†] | 86.76 |
| IWAE + IWAE [1][‡] | 84.78 |
| Stein VAE + ELBO | 85.21 |
| Stein VAE + S-ELBO | 84.98 |
| Stein VIWAE + ELBO | 83.01 |
| Stein VIWAE + S-ELBO | **82.88** |

## 5.2 Density estimation

**Data** We consider five benchmark datasets: MNIST and four text corpora: *20 Newsgroups* (20News), *New York Times* (NYT), *Science* and *RCV1-v2* (RCV2). For MNIST, we used the standard split of 50K training, 10K validation and 10K test examples. The latter three text corpora

consist of 133K, 166K and 794K documents. These three datasets are split into 1K validation, 10K testing and the rest for training.

**Evaluation**   Given new data $\boldsymbol{x}_*$ (testing data), the marginal log-likelihood/perplexity values are estimated by the variational evidence lower bound (ELBO) while integrating the decoder parameters $\boldsymbol{\theta}$ out, $\log p(\boldsymbol{x}_*) \geq \mathbb{E}_{q(\boldsymbol{z}_*)}[\log p(\boldsymbol{x}_*, \boldsymbol{z}_*)] + \mathcal{H}(q(\boldsymbol{z}_*)) = \text{ELBO}(q(\boldsymbol{z}_*))$, where $p(\boldsymbol{x}_*, \boldsymbol{z}_*) = \mathbb{E}_{q(\boldsymbol{\theta})}[\log p(\boldsymbol{x}_*, \boldsymbol{\theta}, \boldsymbol{z}_*)]$ and $\mathcal{H}(q(\cdot)) = -\mathbb{E}_q(\log q(\cdot))$ is the entropy. The expectation is approximated with samples $\{\boldsymbol{\theta}_j\}_{j=1}^M$ and $\{\boldsymbol{z}_{*j}\}_{j=1}^M$ with $\boldsymbol{z}_{*j} = \boldsymbol{f}_{\boldsymbol{\eta}}(\boldsymbol{x}_*, \boldsymbol{\xi}_j), \boldsymbol{\xi}_j \sim q_0(\boldsymbol{\xi})$. Directly evaluating $q(\boldsymbol{z}_*)$ is intractable, thus it is estimated via density transformation $q(\boldsymbol{z}) = q_0(\boldsymbol{\xi})\left|\det\frac{\partial \boldsymbol{f}_{\boldsymbol{\eta}}(\boldsymbol{x}, \boldsymbol{\xi})}{\partial \boldsymbol{\xi}}\right|^{-1}$.

Table 2: Test perplexities on four text corpora.

| Method | 20News | NYT | Science | RCV2 |
|---|---|---|---|---|
| DocNADE [14] | 896 | 2496 | 1725 | 742 |
| DEF [24] | — | 2416 | 1576 | — |
| NVDM [17] | 852 | — | — | 550 |
| Stein VAE + ELBO | 849 | 2402 | 1499 | 549 |
| Stein VAE + S-ELBO | 845 | 2401 | 1497 | 544 |
| Stein VIWAE + ELBO | 837 | 2315 | 1453 | 523 |
| Stein VIWAE + S-ELBO | **829** | **2277** | **1421** | **518** |

We further estimate the marginal log-likelihood/perplexity values via the stochastic variational lower bound, as the mean of 5K-sample importance weighting estimate [1]. Therefore, for each dataset, we report four results: (*i*) *Stein VAE + ELBO*, (*ii*) *Stein VAE + S-ELBO*, (*iii*) *Stein VIWAE + ELBO* and (*iv*) *Stein VIWAE + S-ELBO*; the first term denotes the training procedure is employed as Stein VAE in Section 2 or Stein VIWAE in Section 3; the second term denotes the testing log-likelihood/perplexity is estimated by the ELBO or the stochastic variational lower bound, S-ELBO [1].

**Model**   For MNIST, we train the model with one stochastic layer, $\boldsymbol{z}_n$, with 50 hidden units and two deterministic layers, each with 200 units. The nonlinearity is set as tanh. The visible layer, $\boldsymbol{x}_n$, follows a Bernoulli distribution. For the text corpora, we build a three-layer deep Poisson network [24]. The sizes of hidden units are 200, 200 and 50 for the first, second and third layer, respectively (see [24] for detailed architectures).

**Results**   The log-likelihood/perplexity results are summarized in Tables 1 and 2. On MNIST, our Stein VAE achieves a variational lower bound of -85.21 nats, which outperforms standard VAE with the same model architecture. Our Stein VIWAE achieves a log-likelihood of -82.88 nats, exceeding normalizing flow (-85.1 nats) and importance weighted autoencoder (-84.78 nats), which is the best prior result obtained by feed-forward neural network (FNN). DRAW [5] and PixelRNN [20], which exploit spatial structure, achieved log-likelihoods of around -80 nats. Our model can also be applied on these models, but

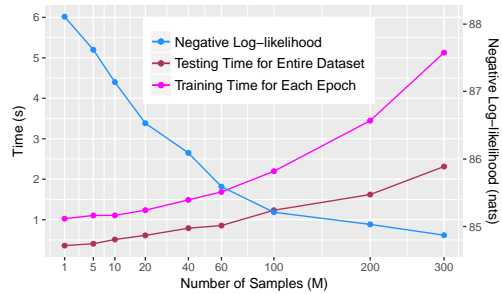

Figure 3: NLL *vs.* Training/Testing time on MNIST with various numbers of samples for $\boldsymbol{\theta}$.

this is left as interesting future work. To further illustrate the benefit of model averaging, we vary the number of samples for $\boldsymbol{\theta}$ (while retaining 100 samples for $\mathcal{Z}$) and show the results associated with training/testing time in Figure 3. When $M = 1$ for $\boldsymbol{\theta}$, our model reduces to a point estimate for that parameter. Increasing the number of samples of $\boldsymbol{\theta}$ (model averaging) improves the negative log-likelihood (NLL). The testing time of using 100 samples of $\boldsymbol{\theta}$ is around 0.12 ms *per image*.

## 5.3   Semi-supervised Classification

We consider semi-supervised classification on MNIST and ImageNet [29] data. For each dataset, we report the results obtained by (*i*) VAE, (*ii*) Stein VAE, and (*iii*) Stein VIWAE.

**MNIST**   We randomly split the training set into a labeled and unlabeled set, and the number of labeled samples in each category varies from 10 to 300. We perform testing on the standard test set with 20 different training-set splits. The decoder for labels is implemented as $p(\boldsymbol{y}_n|\boldsymbol{z}_n, \tilde{\boldsymbol{\theta}}) = \text{softmax}(\tilde{\boldsymbol{\theta}}\boldsymbol{z}_n)$. We consider two types of decoders for images $p(\boldsymbol{x}_n|\boldsymbol{z}_n, \boldsymbol{\theta})$ and encoder $\boldsymbol{f}_{\boldsymbol{\eta}}(\boldsymbol{x}, \boldsymbol{\xi})$:

*(i) FNN*: Following [12], we use a 50-dimensional latent variables $z_n$ and two hidden layers, each with 600 hidden units, for both encoder and decoder; softplus is employed as the nonlinear activation function. *(ii) All convolutional nets (CNN)*: Inspired by [31], we replace the two hidden layers with 32 and 64 kernels of size $5 \times 5$ and a stride of 2. A fully connected layer is stacked on the CNN to produce a 50-dimensional latent variables $z_n$. We use the *leaky* rectified activation [16]. The input of the encoder is formed by spatially aligning and stacking $x_n$ and $\xi$, while the output of decoder is the image itself.

Table 3 shows the classification results. Our Stein VAE and Stein VIWAE consistently achieve better performance than the VAE. We further observe that the variance of Stein VIWAE results is much smaller than that of Stein VAE results on small labeled data, indicating the former produces more robust parameter estimates. State-of-the-art results [26] are achieved by the Ladder network, which can be employed with our Stein-based approach, however, we will consider this extension as future work.

Table 3: Semi-supervised classification error (%) on MNIST. $N_\rho$ is the number of labeled images per class. $\S$[12]; $\dagger$our implementation.

| $N_\rho$ | FNN | | | CNN | | |
|---|---|---|---|---|---|---|
| | VAE$^\S$ | Stein VAE | Stein VIWAE | VAE$^\dagger$ | Stein VAE | Stein VIWAE |
| 10 | $3.33 \pm 0.14$ | $2.78 \pm 0.24$ | $2.67 \pm 0.09$ | $2.44 \pm 0.17$ | $1.94 \pm 0.24$ | $\mathbf{1.90 \pm 0.05}$ |
| 60 | $2.59 \pm 0.05$ | $2.13 \pm 0.08$ | $2.09 \pm 0.03$ | $1.88 \pm 0.05$ | $1.44 \pm 0.04$ | $\mathbf{1.41 \pm 0.02}$ |
| 100 | $2.40 \pm 0.02$ | $1.92 \pm 0.05$ | $1.88 \pm 0.01$ | $1.47 \pm 0.02$ | $1.01 \pm 0.03$ | $\mathbf{0.99 \pm 0.02}$ |
| 300 | $2.18 \pm 0.04$ | $1.77 \pm 0.03$ | $1.75 \pm 0.01$ | $0.98 \pm 0.02$ | $0.89 \pm 0.03$ | $\mathbf{0.86 \pm 0.01}$ |

**ImageNet 2012** We consider scalability of our model to large datasets. We split the 1.3 million training images into an unlabeled and labeled set, and vary the proportion of labeled images from 1% to 40%. The classes are balanced to ensure that no particular class is over-represented, *i.e.*, the ratio of labeled and unlabeled images is the same for each class. We

Table 4: Semi-supervised classification accuracy (%) on ImageNet.

| | VAE | Stein VAE | Stein VIWAE | DGDN [21] |
|---|---|---|---|---|
| 1 % | $35.92 \pm 1.91$ | $36.44 \pm 1.66$ | $36.91 \pm 0.98$ | $\mathbf{43.98 \pm 1.15}$ |
| 2 % | $40.15 \pm 1.52$ | $41.71 \pm 1.14$ | $42.57 \pm 0.84$ | $\mathbf{46.92 \pm 1.11}$ |
| 5 % | $44.27 \pm 1.47$ | $46.14 \pm 1.02$ | $46.20 \pm 0.52$ | $\mathbf{47.36 \pm 0.91}$ |
| 10 % | $46.92 \pm 1.02$ | $47.83 \pm 0.88$ | $\mathbf{48.67 \pm 0.31}$ | $48.41 \pm 0.76$ |
| 20 % | $50.43 \pm 0.41$ | $51.62 \pm 0.24$ | $\mathbf{51.77 \pm 0.12}$ | $51.51 \pm 0.28$ |
| 30 % | $53.24 \pm 0.33$ | $55.02 \pm 0.22$ | $\mathbf{55.45 \pm 0.11}$ | $54.14 \pm 0.12$ |
| 40 % | $56.89 \pm 0.11$ | $58.17 \pm 0.16$ | $\mathbf{58.21 \pm 0.12}$ | $57.34 \pm 0.18$ |

repeat the training process 10 times for the training setting with labeled images ranging from 1% to 10% , and 5 times for the the training setting with labeled images ranging from 20% to 40%. Each time we utilize different sets of images as the unlabeled ones.

We employ an all convolutional net [31] for both the encoder and decoder, which replaces deterministic pooling (*e.g.*, max-pooling) with stridden convolutions. Residual connections [8] are incorporated to encourage gradient flow. The model architecture is detailed in Appendix E. Following [13], images are resized to $256 \times 256$. A $224 \times 224$ crop is randomly sampled from the images or its horizontal flip with the mean subtracted [13]. We set $M = 20$ and $k = 10$.

Table 4 shows classification results indicating that Stein VAE and Stein IVWAE outperform VAE in all the experiments, demonstrating the effectiveness of our approach for semi-supervised classification. When the proportion of labeled examples is too small ($< 10\%$), DGDN [21] outperforms all the VAE-based models, which is not surprising provided that our models are deeper, thus have considerably more parameters than DGDN [21].

## 6 Conclusion

We have employed SVGD to develop a new method for learning a variational autoencoder, in which we need not specify an *a priori* form for the encoder distribution. Fast inference is manifested by learning a recognition model that mimics the manner in which the inferred code samples are manifested. The method is further generalized and improved by performing importance sampling. An extensive set of results, for unsupervised and semi-supervised learning, demonstrate excellent performance and scaling to large datasets.

## Acknowledgements

This research was supported in part by ARO, DARPA, DOE, NGA, ONR and NSF.

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
