[Supplementary Material]

# Appendix for "VAE Learning via Stein Variational Gradient Descent"

**Yunchen Pu, Zhe Gan, Ricardo Henao, Chunyuan Li, Shaobo Han, Lawrence Carin**
Department of Electrical and Computer Engineering, Duke University
{yp42, zg27, r.henao, cl319, shaobo.han, lcarin}@duke.edu

## A  Proof

**Proof of Theorem 1**  Recall the definition of KL divergence:

$$\text{KL}(q_T\|p) = \text{KL}(q_T(\boldsymbol{\theta})q(\mathcal{Z})\|p(\boldsymbol{\theta},\mathcal{Z}|\mathcal{D})) = \int\int q_T(\boldsymbol{\theta})q(\mathcal{Z})\log\frac{p(\boldsymbol{\theta},\mathcal{Z}|\mathcal{D})}{q_T(\boldsymbol{\theta})q(\mathcal{Z})}d\boldsymbol{\theta}d\mathcal{Z} \tag{1}$$

$$= \int q_T(\boldsymbol{\theta})\Big\{\int q(\mathcal{Z})\log p(\boldsymbol{\theta},\mathcal{Z},\mathcal{D})d\mathcal{Z}\Big\}d\boldsymbol{\theta}$$
$$-\int q_T(\boldsymbol{\theta})\log q_T(\boldsymbol{\theta})d\boldsymbol{\theta} - \int q(\mathcal{Z})\log q(\mathcal{Z})d\mathcal{Z} - \log p(\mathcal{D}) \tag{2}$$

$$= \int q_T(\boldsymbol{\theta})\log\tilde{p}(\boldsymbol{\theta};\mathcal{D})d\boldsymbol{\theta} - \int q_T(\boldsymbol{\theta})\log q_T(\boldsymbol{\theta})d\boldsymbol{\theta} - \int q(\mathcal{Z})\log q(\mathcal{Z})d\mathcal{Z} - \log p(\mathcal{D}) \tag{3}$$

$$= \text{KL}\Big(q_T(\boldsymbol{\theta})\|\tilde{p}(\boldsymbol{\theta};\mathcal{D})\Big) - \int q(\mathcal{Z})\log q(\mathcal{Z})d\mathcal{Z} - \log p(\mathcal{D})\,, \tag{4}$$

where $\log\tilde{p}(\boldsymbol{\theta};\mathcal{D}) = \int q(\mathcal{Z})\log p(\boldsymbol{\theta},\mathcal{Z},\mathcal{D})d\mathcal{Z}$. Since $\nabla_\epsilon\int q(\mathcal{Z})\log q(\mathcal{Z})d\mathcal{Z} = \nabla_{\epsilon_1}\log p(\mathcal{D}) = 0$, we have

$$\nabla_\epsilon\text{KL}(q_T(\boldsymbol{\theta})q(\mathcal{Z})\|p(\boldsymbol{\theta},\mathcal{Z}|\mathcal{D})) = \nabla_\epsilon\text{KL}(q_T(\boldsymbol{\theta})\|\tilde{p}(\boldsymbol{\theta};\mathcal{D}))\,. \tag{5}$$

Following [2], we have

$$\nabla_\epsilon(\text{KL}(q_T(\boldsymbol{\theta})q(\mathcal{Z})\|p(\boldsymbol{\theta},\mathcal{Z}|\mathcal{D})|_{\epsilon_1=0}$$
$$= -\mathbb{E}_{\boldsymbol{\theta}\sim q(\theta)}[\nabla_{\boldsymbol{\theta}}\log\tilde{p}(\boldsymbol{\theta};\mathcal{D})^T\psi(\boldsymbol{\theta};\mathcal{D}) + \text{trace}(\nabla_{\boldsymbol{\theta}}\psi(\boldsymbol{\theta};\mathcal{D}))] \tag{6}$$
$$= -\mathbb{E}_{\boldsymbol{\theta}\sim q(\theta)}\Big[\text{trace}\big(\nabla_{\boldsymbol{\theta}}\log\tilde{p}(\boldsymbol{\theta};\mathcal{D})\psi(\boldsymbol{\theta};\mathcal{D})^T + \psi(\boldsymbol{\theta};\mathcal{D}))\big)\Big]\,. \tag{7}$$

**Proof of Theorem 2**  Following [1], we have $\mathbb{E}_{I=\{i_1,\dots,i_m\}}\Big[\frac{1}{m}\sum_{i=j}^m a_{i_j}\Big] = \frac{a_1+\dots+a_k}{k}$, where $I\subset\{1,\dots,k\}$ with $|I| = m < k$, is a uniformly distributed subset of $\{1,\dots,k\}$. Using Jensen's

inequality, we have

$$\mathrm{KL}_{q,p}^{k}(\boldsymbol{\Theta};\mathcal{D}) = -\mathbb{E}_{\boldsymbol{\Theta}^{1:k}\sim q(\boldsymbol{\Theta})}\Big[\log\frac{1}{k}\sum_{i=1}^{k}\frac{p(\boldsymbol{\Theta}^{i}|\mathcal{D})}{q(\boldsymbol{\Theta}^{i})}\Big] \tag{8}$$

$$= -\mathbb{E}_{\boldsymbol{\Theta}^{1:k}\sim q(\boldsymbol{\Theta})}\Big[\log\mathbb{E}_{I=\{i_1,\ldots,i_m\}}\Big[\frac{1}{m}\sum_{i=1}^{m}\frac{p(\boldsymbol{\Theta}^{i}|\mathcal{D})}{q(\boldsymbol{\Theta}^{i})}\Big]\Big] \tag{9}$$

$$\leq -\mathbb{E}_{\boldsymbol{\Theta}^{1:k}\sim q(\boldsymbol{\Theta})}\Big[\mathbb{E}_{I=\{i_1,\ldots,i_m\}}\Big[\log\frac{1}{m}\sum_{i=1}^{m}\frac{p(\boldsymbol{\Theta}^{i}|\mathcal{D})}{q(\boldsymbol{\Theta}^{i})}\Big]\Big] \tag{10}$$

$$= -\mathbb{E}_{\boldsymbol{\Theta}^{1:m}\sim q(\boldsymbol{\Theta})}\Big[\log\frac{1}{m}\sum_{i=1}^{m}\frac{p(\boldsymbol{\Theta}^{i}|\mathcal{D})}{q(\boldsymbol{\Theta}^{i})}\Big] \tag{11}$$

$$= \mathrm{KL}_{q,p}^{m}(\boldsymbol{\Theta};\mathcal{D}), \tag{12}$$

if $q(\boldsymbol{\Theta})/p(\boldsymbol{\Theta}|\mathcal{D})$ is bounded, we have

$$\lim_{k\to\infty}\frac{1}{k}\sum_{i=1}^{k}\frac{p(\boldsymbol{\Theta}^{i}|\mathcal{D})}{q(\boldsymbol{\Theta}^{i})} = \mathbb{E}_{q(\boldsymbol{\Theta})}\Big[\frac{p(\boldsymbol{\Theta}|\mathcal{D})}{q(\boldsymbol{\Theta})}\Big] = \int p(\boldsymbol{\Theta}|\mathcal{D})d\boldsymbol{\Theta} = 1. \tag{13}$$

Therefore

$$\mathrm{KL}_{q,p}^{k}(\boldsymbol{\Theta};\mathcal{D}) = -\lim_{k\to\infty}\mathbb{E}_{\boldsymbol{\Theta}^{1:k}\sim q(\boldsymbol{\Theta})}\Big[\log 1\Big] = 0.$$

**Proof of Theorem 3** $\mathcal{A}_{p}^{k}(\boldsymbol{\Theta}^{1:k};\mathcal{D})$ is defined as following:

$$\mathcal{A}_{p}^{k}(\boldsymbol{\Theta}^{1:k};\mathcal{D}) = \frac{1}{\tilde{\omega}}\sum_{i=1}^{k}\omega_{i}\Big(\mathrm{trace}\big(\mathcal{A}_{p}(\boldsymbol{\Theta}^{i};\mathcal{D})\big)\Big) \tag{14}$$

$$\omega_{i} = p(\boldsymbol{\Theta}^{i};\mathcal{D})/q(\boldsymbol{\Theta}^{i}), \quad \tilde{\omega} = \sum_{i=1}^{k}\omega_{i} \tag{15}$$

$$\mathcal{A}_{p}(\boldsymbol{\Theta};\mathcal{D}) = \nabla_{\boldsymbol{\Theta}}\log\tilde{p}(\boldsymbol{\Theta};\mathcal{D})\psi(\boldsymbol{\Theta};\mathcal{D})^{T} + \nabla_{\boldsymbol{\Theta}}\psi(\boldsymbol{\Theta};\mathcal{D}). \tag{16}$$

Assume $p_{[T^{-1}]}(\boldsymbol{\Theta})$ denote the density of $\hat{\boldsymbol{\Theta}} = T^{-1}(\boldsymbol{\Theta})$. We have

$$\nabla_{\epsilon}\Big(\mathrm{KL}_{q,p}^{k}(\boldsymbol{\Theta}';\mathcal{D})\Big) = -\nabla_{\epsilon}\Big\{\mathbb{E}_{\boldsymbol{\Theta}^{1:k}\sim q(\boldsymbol{\Theta})}\Big[\log\frac{1}{k}\sum_{i=1}^{k}\frac{p_{[T^{-1}]}(\boldsymbol{\Theta}^{i}|\mathcal{D})}{q(\boldsymbol{\Theta}^{i})}\Big]\Big\} \tag{17}$$

$$= -\mathbb{E}_{\boldsymbol{\Theta}^{1:k}\sim q(\boldsymbol{\Theta})}\Big\{\nabla_{\epsilon}\Big[\log\frac{1}{k}\sum_{i=1}^{k}\frac{p_{[T^{-1}]}(\boldsymbol{\Theta}^{i}|\mathcal{D})}{q(\boldsymbol{\Theta}^{i})}\Big]\Big\} \tag{18}$$

$$= -\mathbb{E}_{\boldsymbol{\Theta}^{1:k}\sim q(\boldsymbol{\Theta})}\Big\{\Big[\frac{1}{k}\sum_{i=1}^{k}\frac{p_{[T^{-1}]}(\boldsymbol{\Theta}^{i}|\mathcal{D})}{q(\boldsymbol{\Theta}^{i})}\Big]^{-1}\Big[\frac{1}{k}\sum_{i=1}^{k}\frac{\nabla_{\epsilon}p_{[T^{-1}]}(\boldsymbol{\Theta}^{i}|\mathcal{D})}{q(\boldsymbol{\Theta}^{i})}\Big]\Big\}. \tag{19}$$

Note that

$$\nabla_{\epsilon}p_{[T^{-1}]}(\boldsymbol{\Theta}^{i}|\mathcal{D}) = p_{[T^{-1}]}(\boldsymbol{\Theta}^{i}|\mathcal{D})\nabla_{\epsilon}\log p_{[T^{-1}]}(\boldsymbol{\Theta}^{i}|\mathcal{D}), \tag{20}$$

and when $\epsilon = 0$, we have

$$p_{[T^{-1}]}(\boldsymbol{\Theta}^{i}|\mathcal{D}) = p(\boldsymbol{\Theta}^{i}|\mathcal{D}), \qquad\qquad \nabla_{\epsilon}T(\boldsymbol{\Theta}) = \psi(\boldsymbol{\Theta};\mathcal{D}) \tag{21}$$

$$\nabla_{\epsilon}\nabla_{\boldsymbol{\Theta}}T(\boldsymbol{\Theta}) = \nabla_{\epsilon}\psi(\boldsymbol{\Theta};\mathcal{D}), \qquad\qquad \nabla_{\boldsymbol{\Theta}}T(\boldsymbol{\Theta}) = \mathbf{I} \tag{22}$$

Therefore

$$\nabla_{\epsilon}\log p_{[T^{-1}]}(\boldsymbol{\Theta}^{i}|\mathcal{D}) = \nabla_{\epsilon}\log p(\boldsymbol{\Theta}^{i}|\mathcal{D})^{T}\nabla_{\epsilon}T(\boldsymbol{\Theta}^{i}) + \mathrm{trace}\Big(\big(\nabla_{\boldsymbol{\Theta}^{i}}T(\boldsymbol{\Theta}^{i})\big)^{-1}\cdot\nabla_{\epsilon}\nabla_{\boldsymbol{\Theta}^{i}}T(\boldsymbol{\Theta}^{i})\Big)$$

$$= \nabla_{\epsilon}\log p(\boldsymbol{\Theta}^{i}|\mathcal{D})^{T}\psi(\boldsymbol{\Theta}^{i};\mathcal{D}) + \mathrm{trace}\big(\nabla_{\epsilon}\psi(\boldsymbol{\Theta}^{i};\mathcal{D})\big) \tag{23}$$

$$= \mathrm{trace}\big(\nabla_{\epsilon}\log p(\boldsymbol{\Theta}^{i}|\mathcal{D})\psi(\boldsymbol{\Theta}^{i};\mathcal{D})^{T} + \nabla_{\epsilon}\psi(\boldsymbol{\Theta}^{i};\mathcal{D})\big) \tag{24}$$

$$= \mathrm{trace}\big(\mathcal{A}_{p}(\boldsymbol{\Theta}^{i};\mathcal{D})\big). \tag{25}$$

Therefore, (19) can be rewritten as

$$
\begin{aligned}
\nabla_\epsilon\Big(\mathrm{KL}_{q,p}^k(\boldsymbol{\Theta}';\mathcal{D})\Big) &= -\mathbb{E}_{\boldsymbol{\Theta}^{1:k}\sim q(\boldsymbol{\Theta})}\left\{\Big[\frac{1}{k}\sum_{i=1}^{k}\frac{p_{[T^{-1}]}(\boldsymbol{\Theta}^i|\mathcal{D})}{q(\boldsymbol{\Theta}^i)}\Big]^{-1}\Big[\frac{1}{k}\sum_{i=1}^{k}\frac{\nabla_\epsilon p_{[T^{-1}]}(\boldsymbol{\Theta}^i|\mathcal{D})}{q(\boldsymbol{\Theta}^i)}\Big]\right\}\\
&= -\mathbb{E}_{\boldsymbol{\Theta}^{1:k}\sim q(\boldsymbol{\Theta})}\left\{\Big[\sum_{i=1}^{k}\frac{p(\boldsymbol{\Theta}^i|\mathcal{D})}{q(\boldsymbol{\Theta}^i)}\Big]^{-1}\Big[\sum_{i=1}^{k}\frac{p(\boldsymbol{\Theta}^i|\mathcal{D})}{q(\boldsymbol{\Theta}^i)}\nabla_\epsilon\log p_{[T^{-1}]}(\boldsymbol{\Theta}^i|\mathcal{D})\Big]\right\}\\
&= -\mathbb{E}_{\boldsymbol{\Theta}^{1:k}\sim q(\boldsymbol{\Theta})}\left\{\frac{1}{\tilde{\omega}}\sum_{i=1}^{k}\omega_i\Big[\mathrm{trace}\big(\mathcal{A}_p(\boldsymbol{\Theta}^i;\mathcal{D})\big)\Big]\right\},
\end{aligned}
\tag{26}
$$

where $\omega_k = p(\boldsymbol{\Theta}^i;\mathcal{D})/q(\boldsymbol{\Theta}^i)$ and $\tilde{\omega}=\sum_{i=1}^{k}\omega_i$.

# B    Samples Updating for Stein VIWAE

let $\{\boldsymbol{\theta}_j^{1:k,t}\}_{j=1}^{M}$ and $\{\boldsymbol{z}_{jn}^{1:k,t}\}_{j=1}^{M}$ denote the samples acquired at iteration $t$ of the learning procedure. To update samples of $\boldsymbol{\theta}^{1:k}$, we apply the transformation $\boldsymbol{\theta}_j^{(i,t+1)} = T(\boldsymbol{\theta}_j^{(i,t)};\mathcal{D}) = \boldsymbol{\theta}_j^{(i,t)} + \epsilon\psi(\boldsymbol{\theta}_j^{(i,t)};\mathcal{D})$, for $i=1,\dots,k$, by approximating the expectation by samples $\{\boldsymbol{z}_{jn}^{1:k}\}_{j=1}^{M}$, and we have

$$
\boldsymbol{\theta}_j^{(i,t+1)} = \boldsymbol{\theta}_j^{(i,t)} + \epsilon\Delta\boldsymbol{\theta}_j^{(i,t)}, \text{ for } i=1,\dots,k,
\tag{27}
$$

with

$$
\Delta\boldsymbol{\theta}_j^{(i,t)} \approx \frac{1}{M}\sum_{j'=1}^{M}\left[\frac{1}{\tilde{\omega}}\sum_{i'=1}^{k}\omega_i\big(\nabla_{\boldsymbol{\theta}_{j'}^{(i',t)}}k_{\boldsymbol{\theta}}(\boldsymbol{\theta}_{j'}^{(i',t)},\boldsymbol{\theta}_j^{(i,t)})\big) + k(\boldsymbol{\theta}_{j'}^{(i',t)},\boldsymbol{\theta}_j^{(i,t)})\nabla_{\boldsymbol{\theta}_{j'}^{(i',t)}}\log\tilde{p}(\boldsymbol{\theta}_{j'}^{(i',t)};\mathcal{D})\right]
\tag{28}
$$

$$
\omega_i \approx \frac{1}{M}\sum_{n=1}^{N}\sum_{j=1}^{M}\frac{p(\boldsymbol{\theta}^i,\boldsymbol{z}_{jn}^i,\boldsymbol{x}_n)}{q(\boldsymbol{\theta}^i)q(\boldsymbol{z}_{jn}^i)},\ \tilde{\omega}=\sum_{i=1}^{k}\omega_i
\tag{29}
$$

$$
\nabla_{\boldsymbol{\theta}}\log\tilde{p}(\boldsymbol{\theta};\mathcal{D}) \approx \frac{1}{M}\sum_{n=1}^{N}\sum_{j=1}^{M}\nabla_{\boldsymbol{\theta}}\log p(\boldsymbol{x}_n|\boldsymbol{z}_{jn},\boldsymbol{\theta})p(\boldsymbol{\theta}).
$$

Similarly, when updating samples of the latent variables, we have

$$
\boldsymbol{z}_{jn}^{(i,t+1)} = \boldsymbol{z}_{jn}^{(i,t)} + \epsilon\Delta\boldsymbol{z}_{jn}^{(i,t)}, \text{ for } i=1,\dots,k,
\tag{30}
$$

with

$$
\Delta\boldsymbol{z}_{jn}^{(i,t)} \approx \frac{1}{M}\sum_{j'=1}^{M}\left[\frac{1}{\tilde{\omega}_n}\sum_{i'=1}^{k}\omega_{in}\big(\nabla_{\boldsymbol{z}_{j'n}^{(i',t)}}k_{\boldsymbol{z}}(\boldsymbol{z}_{j'n}^{(i',t)},\boldsymbol{z}_{jn}^{(i,t)})\big) + k_{\boldsymbol{z}}(\boldsymbol{z}_{j'n}^{(i',t)},\boldsymbol{z}_{jn}^{(i,t)})\nabla_{\boldsymbol{z}_{j'n}^{(i',t)}}\log\tilde{p}(\boldsymbol{z}_{j'n}^{(i',t)};\mathcal{D})\right]
\tag{31}
$$

$$
\omega_{in} \approx \frac{1}{M}\sum_{j=1}^{M}\frac{p(\boldsymbol{\theta}^i,\boldsymbol{z}_{jn}^i,\boldsymbol{x}_n)}{q(\boldsymbol{\theta}^i)q(\boldsymbol{z}_{jn}^i)},\ \tilde{\omega}_n=\sum_{i=1}^{k}\omega_{in}
\tag{32}
$$

$$
\nabla_{\boldsymbol{z}_n}\log\tilde{p}(\boldsymbol{z}_n;\mathcal{D}) \approx \frac{1}{M}\sum_{j=1}^{M}\nabla_{\boldsymbol{z}_n}\log p(\boldsymbol{x}_n|\boldsymbol{z}_n,\boldsymbol{\theta}_j')p(\boldsymbol{z}_n)
\tag{33}
$$

# C    Samples Updating for Semi-supervised Learning

The expectations in (13) and (14) in the main paper are approximated by samples. For updating samples of $\boldsymbol{\theta}$, we have

$$
\boldsymbol{\theta}_j^{(t+1)} = \boldsymbol{\theta}_j^{(t)} + \epsilon_1\Delta\boldsymbol{\theta}_j^{(t)},
\tag{34}
$$

with

$$\Delta\boldsymbol{\theta}_j^{(t)} \approx \frac{1}{M}\sum_{j'=1}^{M}[k_{\boldsymbol{\theta}}(\boldsymbol{\theta}_{j'}^{(t)},\boldsymbol{\theta}_j^{(t)})\nabla_{\boldsymbol{\theta}_{j'}^{(t)}}\log\tilde{p}(\boldsymbol{\theta}_{j'}^{(t)};\mathcal{D},\mathcal{D}_l)e + \nabla_{\boldsymbol{\theta}_{j'}^{(t)}}k_{\boldsymbol{\theta}}(\boldsymbol{\theta}_{j'}^{(t)},\boldsymbol{\theta}_j^{(t)}))] \tag{35}$$

$$\nabla_{\boldsymbol{\theta}}\log\tilde{p}(\boldsymbol{\theta};\mathcal{D},\mathcal{D}_l) \approx \frac{1}{M}\sum_{j=1}^{M}\Big\{\sum_{\boldsymbol{x}_n\in\mathcal{D}}\nabla_{\boldsymbol{\theta}}\log p(\boldsymbol{x}_n|\boldsymbol{z}_{jn},\boldsymbol{\theta}) + \sum_{\boldsymbol{x}_n\in\mathcal{D}_l}\nabla_{\boldsymbol{\theta}}\log p(\boldsymbol{x}_n|\boldsymbol{z}_{jn},\boldsymbol{\theta})\Big\}p(\boldsymbol{\theta})\,. \tag{36}$$

Similarly, when updating samples of $\tilde{\boldsymbol{\theta}}$ , we have

$$\tilde{\boldsymbol{\theta}}_j^{(t+1)} = \tilde{\boldsymbol{\theta}}_j^{(t)} + \epsilon_2\Delta\tilde{\boldsymbol{\theta}}_j^{(t)}\,, \tag{37}$$

with

$$\Delta\tilde{\boldsymbol{\theta}}_j^{(t)} \approx \frac{1}{M}\sum_{j'=1}^{M}[k_{\tilde{\boldsymbol{\theta}}}(\tilde{\boldsymbol{\theta}}_{j'}^{(t)},\tilde{\boldsymbol{\theta}}_j^{(t)})\nabla_{\tilde{\boldsymbol{\theta}}_{j'}^{(t)}}\log\tilde{p}(\tilde{\boldsymbol{\theta}}_{j'}^{(t)};\mathcal{D}_l) + \nabla_{\tilde{\boldsymbol{\theta}}_{j'}^{(t)}}k_{\tilde{\boldsymbol{\theta}}}(\tilde{\boldsymbol{\theta}}_{j'}^{(t)},\tilde{\boldsymbol{\theta}}_j^{(t)}))]$$

$$\nabla_{\tilde{\boldsymbol{\theta}}}\log\tilde{p}(\tilde{\boldsymbol{\theta}};\mathcal{D}_l) \approx \frac{1}{M}\sum_{j=1}^{M}\sum_{\boldsymbol{y}_n\in\mathcal{D}_l}\nabla_{\tilde{\boldsymbol{\theta}}}\log p(\boldsymbol{y}_n|\boldsymbol{z}_{jn},\tilde{\boldsymbol{\theta}})p(\tilde{\boldsymbol{\theta}})\,. \tag{38}$$

Similarly, samples of $\boldsymbol{z}_n\in\mathcal{Z}_l$ are updated

$$\boldsymbol{z}_{jn}^{(t+1)} = \boldsymbol{z}_{jn}^{(t)} + \epsilon\Delta\boldsymbol{z}_{jn}^{(t)}\,, \tag{39}$$

with

$$\Delta\boldsymbol{z}_{jn}^{(t)} = \frac{1}{M}\sum_{j'=1}^{M}[k_{\boldsymbol{z}}(\boldsymbol{z}_{j'n}^{(t)},\boldsymbol{z}_{jn}^{(t)})\nabla_{\boldsymbol{z}_{j'n}^{(t)}}\log\tilde{p}(\boldsymbol{z}_{j'n}^{(t)};\mathcal{D}_l) + \nabla_{\boldsymbol{z}_{j'n}^{(t)}}k_{\boldsymbol{z}}(\boldsymbol{z}_{j'n}^{(t)},\boldsymbol{z}_{jn}^{(t)}))] \tag{40}$$

$$\nabla_{\boldsymbol{z}_n}\log\tilde{p}(\boldsymbol{z}_n;\mathcal{D}_l) \approx \frac{1}{M}\sum_{j=1}^{M}\nabla_{\boldsymbol{z}_n}p(\boldsymbol{z}_n)\Big\{\log p(\boldsymbol{x}_n|\boldsymbol{z}_n,\boldsymbol{\theta}_j') + \zeta\log p(\boldsymbol{y}_n|\boldsymbol{z}_n,\tilde{\boldsymbol{\theta}}_j')\Big\}\,, \tag{41}$$

where $\zeta$ is a tuning parameter that balances the two components. Motivated by assigning the same weight to every data point [3], we set $\zeta = N_X/(C\rho)$ in the experiments, where $N_X$ is the dimension of $\boldsymbol{x}_n$, $C$ is the number of categories for the corresponding label and $\rho$ is the proportion of labeled data in the mini-batch.

## D   Posterior of Gaussian Mixture Model

Consider $\boldsymbol{z}\sim\frac{1}{2}\mathcal{N}(\boldsymbol{\mu}_1,\mathbf{I}) + \frac{1}{2}\mathcal{N}(\boldsymbol{\mu}_2,\mathbf{I})$ and $\boldsymbol{x}_n\sim\mathcal{N}(\boldsymbol{\theta}\boldsymbol{z},\sigma^2\mathbf{I})$, where $\boldsymbol{z}\in\mathbb{R}^K$, $\boldsymbol{x}\in R^P$ and $\boldsymbol{\theta}\in\mathbb{R}^{P\times K}$. We have

$$p(\boldsymbol{z}|\boldsymbol{x}) \propto p(\boldsymbol{x})p(\boldsymbol{z}) \propto \exp\Big\{-\frac{(\boldsymbol{x}-\boldsymbol{\theta}\boldsymbol{z})^T(\boldsymbol{x}-\boldsymbol{\theta}\boldsymbol{z})}{2\sigma^2}\Big\}$$

$$\times\Big\{\exp\Big\{-\frac{(\boldsymbol{z}-\boldsymbol{\mu}_1)^T(\boldsymbol{z}-\boldsymbol{\mu}_1)}{2}\Big\} + \exp\Big\{-\frac{(\boldsymbol{z}-\boldsymbol{\mu}_2)^T(\boldsymbol{z}-\boldsymbol{\mu}_2)}{2}\Big\}\Big\} \tag{42}$$

$$= \exp\Big\{-\frac{1}{2}\Big[\boldsymbol{z}^T\big(\frac{\boldsymbol{\theta}^T\boldsymbol{\theta}}{\sigma^2}+\mathbf{I}\big)\boldsymbol{z} - 2\big(\frac{\boldsymbol{y}^T\boldsymbol{\theta}}{\sigma^2}+\boldsymbol{\mu}_1\big)\boldsymbol{z} + \frac{\boldsymbol{x}^T\boldsymbol{x}}{\sigma^2}+\boldsymbol{\mu}_1^T\boldsymbol{\mu}_1\Big]\Big\}$$

$$+ \exp\Big\{-\frac{1}{2}\Big[\boldsymbol{z}^T\big(\frac{\boldsymbol{\theta}^T\boldsymbol{\theta}}{\sigma^2}+\mathbf{I}\big)\boldsymbol{z} - 2\big(\frac{\boldsymbol{y}^T\boldsymbol{\theta}}{\sigma^2}+\boldsymbol{\mu}_2\big)\boldsymbol{z} + \frac{\boldsymbol{x}^T\boldsymbol{x}}{\sigma^2}+\boldsymbol{\mu}_2^T\boldsymbol{\mu}_2\Big]\Big\}\,. \tag{43}$$

Let

$$\boldsymbol{\Sigma} = \frac{\boldsymbol{\theta}^T\boldsymbol{\theta}}{\sigma^2}+\mathbf{I}, \quad \hat{\boldsymbol{\mu}}_1 = \boldsymbol{\Sigma}^{-1}\big(\frac{\boldsymbol{y}^T\boldsymbol{\theta}}{\sigma^2}-\boldsymbol{\mu}_1\big), \qquad p_1 = \frac{\boldsymbol{x}^T\boldsymbol{x}}{\sigma^2}+\boldsymbol{\mu}_1^T\boldsymbol{\mu}_1 - \hat{\boldsymbol{\mu}}_1^T\boldsymbol{\Sigma}\hat{\boldsymbol{\mu}}_1, \tag{44}$$

$$\hat{\boldsymbol{\mu}}_2 = \boldsymbol{\Sigma}^{-1}\big(\frac{\boldsymbol{y}^T\boldsymbol{\theta}}{\sigma^2}-\boldsymbol{\mu}_2\big), \qquad p_2 = \frac{\boldsymbol{x}^T\boldsymbol{x}}{\sigma^2}+\boldsymbol{\mu}_2^T\boldsymbol{\mu}_2 - \hat{\boldsymbol{\mu}}_2^T\boldsymbol{\Sigma}\hat{\boldsymbol{\mu}}_2\,, \tag{45}$$

The density in (43) can be rewritten as

$$p(\boldsymbol{z}|\boldsymbol{x}) \propto \exp\{p_1\} \exp\left\{-\frac{1}{2}(\boldsymbol{z}-\hat{\boldsymbol{\mu}}_1)^T \boldsymbol{\Sigma}(\boldsymbol{z}-\hat{\boldsymbol{\mu}}_1)\right\} + \exp\{p_2\} \exp\left\{-\frac{1}{2}(\boldsymbol{z}-\hat{\boldsymbol{\mu}}_2)^T \boldsymbol{\Sigma}(\boldsymbol{z}-\hat{\boldsymbol{\mu}}_2)\right\}.$$

(46)

Therefore, we have $\boldsymbol{z}|\boldsymbol{x} \sim p(\boldsymbol{z}|\boldsymbol{x}) = \hat{p}\mathcal{N}(\hat{\boldsymbol{\mu}}_1, \boldsymbol{\Sigma}) + (1-\hat{p})\mathcal{N}(\hat{\boldsymbol{\mu}}_2, \boldsymbol{\Sigma})$, where

$$\hat{p} = \frac{1}{1+\exp(p_2 - p_1)}.$$

(47)

# E   Model Architecture

Table 1: Architecture of the models for semi-supervised classification on ImageNet. BN denotes batch normalization. The layer in bracket indicates the number of layers stacked.

| Output Size | Encoder | Decoder |
|---|---|---|
| $224 \times 224 \times 4$ for encoder<br>$224 \times 224 \times 3$ for decoder | RGB image $\boldsymbol{x}_n$ stacked by $\boldsymbol{\xi}$ | RGB image $\boldsymbol{x}_n$ |
| $56 \times 56 \times 64$ | $7 \times 7$ conv, 64 kernels, LeakyRelu, stride 4, BN<br>$\left[ 3 \times 3 \text{ conv, 64 kernels, LeakyRelu, stride 1, BN} \right] \times 3$ | |
| $28 \times 28 \times 128$ | $3 \times 3$ conv, 128 kernels, LeakyRelu, stride 2, BN<br>$\left[ 3 \times 3 \text{ conv, 128 kernels, LeakyRelu, stride 1, BN} \right] \times 3$ | |
| $14 \times 14 \times 256$ | $3 \times 3$ conv, 256 kernels, LeakyRelu, stride 2, BN<br>$\left[ 3 \times 3 \text{ conv, 256 kernels, LeakyRelu, stride 1, BN} \right] \times 3$ | |
| $7 \times 7 \times 512$ | $3 \times 3$ conv, 512 kernels, LeakyRelu, stride 2, BN<br>$\left[ 3 \times 3 \text{ conv, 512 kernels, LeakyRelu, stride 1, BN} \right] \times 3$ | |
| latent code $\boldsymbol{z}_n$ | | |
| $1 \times 1$ conv, 2048 kernels, LeakyRelu | | |
| average pooling, 1000-dimensional fully connected layer | | |
| softmax, label $\boldsymbol{y}_n$ | | |

# F    Additional Results

**Gaussian Mixture Model**    Figure 1 and 2 show the performance of Stein VAE approximations for the true posterior using $M = 10$, $M = 20$, $M = 50$ and $M = 100$ samples on test data.

(a) $M = 10$          (b) $M = 20$          (c) $M = 50$          (d) $M = 100$

Figure 1: Approximation of posterior distribution: Stein VAE *vs.* VAE. The figures represent different samples of Stein VAE. Each row corresponds to the same test data, and each column corresponds to the same number of samples with (a) 10 samples; (b) 20 samples; (c) 50 samples; (d) 100 samples.

Figure 2: Approximation of posterior distribution: Stein VAE *vs.* VAE. The figures represent different samples of Stein VAE. Each row corresponds to the same test data, and each column corresponds to the same number of samples with (a) 10 samples; (b) 20 samples; (c) 50 samples; (d) 100 samples.

**Poisson Factor Analysis**    We show the marginal and pairwise posteriors of test data in Figure 3.

Figure 3: Univariate marginals and pairwise posteriors. Purple, red and green represent the distribution inferred from MCMC, standard VAE and Stein VAE, respectively.