[Reviews · NeurIPS 2017]

Reviewer 1



%%% UPDATE: Thank you for your response, which has been read %%% %%% The score I gave has increased by one level, after I was assured that this work is of some interest to research on VAE %%% %%% Please ensure that ALL details needed to reproduce the experiments are included in any revision of the paper %%% This paper contains a lot of different ideas around variational auto-encoders and Stein's method, which are brought together in a way that appears novel. However I feel that the presentation could be improved and more details about the (complicated) implementation should have been included. - The background on variational auto-encoders was not clear, to this reviewer who has not worked with VAEs. For instance, first p(x | z_n ; \theta) is called the "encoder", then q(z | x ; \phi) is called the "encoder" - which is it, and why is it given this name? - \top should be used as transpose in latex. - What is the domain of \theta? Related, how is \grad_\theta \psi(\theta ; D) defined if both \theta and \psi are vectors? - Several approximations are composed en route to the final method that is proposed. It is unclear how all these approximations interact - this isn't assessed theoretically - so that all of the assessment must be based on the empirical results. - Unfortunately several details necessary to reproduce the results seem to have been omitted. e.g. the value of \epsilon (step size for SVGD), the number of SVGD iterations (or the stopping criterion), the initial locations of the particles, etc. In particular, the initial location of the particles would presumably be quite influential to the performance of SVGD. - I did not understand Theorem 2. The claim seems to be that KL_{q,p}^k is involved in a tighter variational lower bound, but what is the significance of the fact that Lim_{k \rightarrow \infty} KL_{q,p}^k(\Theta ; D) = 0? This is not really explained. - On line 178, "q_0" has not been defined, I think. - In general there are a lot of moving parts in the proposed method, and most of these "levers" are unexplored in the experiments. For instance, how does the choice of k impact on performance? - An MCMC benchmark is used for a couple of the experiments - there is presumably an implicit argument that the proposed method is faster than MCMC. However, this isn't actually tested and computational times against MCMC are not included.

Reviewer 2



Recent work has shown that the kernel Stein discrepancy can be viewed in the context of gradient flows on distributions, i.e., the optimal test function of the kernel Stein discrepancy is also a type of functional gradient of a perturbed distribution w.r.t. the KL divergence [https://arxiv.org/abs/1608.04471]. The main idea in this work is to use this gradient flow property to improve the training of a variational autoencoder. Instead of using backprop on a single sample at a time, the authors show one can use this gradient flow property of the kernel Stein discrepancy to backprop w.r.t. to an entire sample, which substantially reduces the variance of the gradient that regularizes the latent space toward the marginal distribution of p(theta, z). The authors also implement these ideas with those of importance weighted VAEs, and accordingly achieve better results than the vanilla VAEs. Since these ideas are quite often orthogonal to other ideas used to improve VAEs, this seems like a key ingredient to obtain state-of-the-art VAEs and a worthy contribution to the literature. Detailed Comments: L38: Its a bit awkward to call p(x|z) and q(x|z) the encoder (for q(x|z) it seems natural but p(x|z) is just a conditional distribution). L84: Are these z_{jn} supposed to be z_{jn}^{t-1}? Or are they really all drawn from q(Z) still as suggested in Theorem 1? L86: Same as before. Also, shouldn't this be theta_{j'} and not theta_j'? L178: What's q0(theta)? Is this supposed to be q(theta)? If so whats q(Z)? My understanding is these are how you draw the initial theta and z samples, correct? If so, I've noticed that many of the following examples leave the prior p(theta, z) unspecificed... L186-190: Are theta the parameters to be learned here? If so, what's the prior p(theta) on theta? If the statement before is incorrect, what's the form of the decoder? This is confusing. L187: How many layers are in the MLP? Just one layer of hidden units? L216: How many layers in the MLP?

Reviewer 3



This well-written paper presents a method for training variational autoencoders by performing alternating updates on the codes Z and parameters theta using Stein variational gradient descent (SVGD). SVGD was previously proposed as a general-purpose algorithm for variational inference but had not yet been applied to autoencoders prior to this paper. A rich set of experiments is presented, showing that the Stein VAE outperforms the standard VAE in recognizing multimodal posteriors and accurately capturing posterior variance rather than underestimating it. I wish some timing comparisons had been provided for training the Stein VAE versus the standard VAE, but this is a minor comment. The experiments are certainly far more compelling than those in the original SVGD paper.